# Structure and Properties of Bioactive Glass-Modified Calcium Phosphate/Calcium Sulfate Biphasic Porous Self-Curing Bone Repair Materials and Preliminary Research on Their Osteogenic Effect

**DOI:** 10.3390/ma15227898

**Published:** 2022-11-08

**Authors:** Tao Tan, Danyang Song, Suning Hu, Xiangrui Li, Mei Li, Lei Wang, Hailan Feng

**Affiliations:** 1Department of Prosthodontics, Peking University School and Hospital of Stomatology, National Clinical Research Center for Oral Diseases, National Engineering Research Center of Oral Biomaterials and Digital Medical Devices, Beijing Key Laboratory of Digital Stomatology, Haidian District, Beijing 100081, China; 2Dental Clinic, Peking University International Hospital, Life Garden Road, Changping District, Beijing 102206, China; 3Beijing Naton Medical Institute Co., Ltd., Building 1, Yard 9, Chengwan Street, Haidian District, Beijing 100086, China

**Keywords:** bone repair materials, scanning electron microscopy (SEM), compressive strength, osteogenesis

## Abstract

In this study, calcium phosphate (CP)/calcium sulfate biphasic bone repair materials were modified with bioactive-glass (BG) to construct a self-curing bone repair material. Tetracalcium phosphate, calcium hydrogen phosphate dihydrate, and calcium sulfate hemihydrate (CSH) with different BG ratios and phosphate solution were reacted to prepare a porous self-curing bone repair material (CP/CSH/BG). The solidification time was about 12 min, and the material was morphologically stable in 24 h. The porosity was about 50%, with a pore size around 200 μm. The strength of CP/CSH/BG was approaching trabecular bone, and could be gradually degraded in Tris-HCl solution. MC3T3-E1 cells were cultured in the leaching solution of the materials. Cytotoxicity was detected using Cell Counting Kit 8 assays, and the expression of osteogenesis-related biomarkers was detected using quantitative real-time reverse transcription PCR (qRT-PCR). The results showed that all BG groups had increased ALP and ARS staining, implying that the BG groups could promote osteoblast mineralization in vitro. qRT-PCR showed significant upregulation of bone-related gene expression (*Osx*, *Ocn*, *Runx2*, and *Col1*) in the 20% BG group (*p* < 0.05). Therefore, the CP/CSH/BG self-curing bone repair materials can promote osteogenesis, and might be applied for bone regeneration, especially for polymorphic bone defect repair.

## 1. Introduction

Oral and maxillofacial bone defects are often caused by tumors, trauma, and periodontal disease, which make the defects more polymorphic and, thus, more challenging to treat. Currently, four sources of bone repair materials are used: autogenous, allogeneic, xenogeneic, and artificial materials [1]. Among them, autogenous bone has the best therapeutic effect but involves complex manipulation for bone harvesting and trimming to fit the recipient region, and has potential complications in donor side morbidities such as pain, infection, hematoma, swelling, fracture, visceral complications, and paresthesia [2]. Allogeneic and xenogeneic materials have a natural micropore structure that is conducive to osteogenesis; however, the shaping required to fit them to recipients is time-consuming and technically demanding [3]. Compared with the first three materials, artificial bone has the advantages of being manufactured on demand with the correct structure and volume. In particular, artificial bone can be injected and shaped easily to the size of the defect, thus achieving minimally invasive insertion and self-curing, making bone repair manipulation more convenient [4].

The use of a composite bone repair material containing calcium phosphate (CP) and calcium sulfate (CS) as an artificial material has been described in several reports [5,6], which demonstrated favorable healing outcomes. However, they generally lack suitable porosity for cell migration and tissue formation [7]. Bioactive glass (BG) is biologically active in bone formation due to its osteo-inductive effects [8]. A variety of BG materials have been used clinically to repair bone defects in non load-bearing regions. Moreover, its pores can promote the establishment of a capillary network, thus promoting the colonization of osteoblasts and the transfer of nutrients [9,10].

In this study, CP/CS was used to synthesize injectable composite bone substitute materials. Different proportions of BG were used as modifiers to increase the porosity and improve the biological activity of the composite material. Meanwhile, different pore forming methods were used to increase the porosity of the composite material. The degradability, mechanical properties, anti-collapsibility, and microstructure of the CP/CSH/BG materials were observed and evaluated, and their osteogenic properties were evaluated in vitro.

## 2. Materials and Methods

### 2.1. Material Preparation and Porous Structure Formation

All the reagents are commercially available, including calcium hydrogen phosphate dihydrate (Zhanwang Pharmaceutical, Huzhou, China), Tetracalcium phosphate (TTCP), and BG (Naton Medical Group, Beijing, China), calcium sulfate hemihydrate (CSH) (Doman Pharmaceutical Co., Ltd., Shanghai, China), and Rebone bone repair material (Rebone Bio, Shanghai, China).

TTCP and dicalcium phosphate (DCPD) were mixed in the same proportion, both of which accounted for 50 wt%, and then different proportions of (0–30 wt%) BG and (50–20 wt%) CSH were incorporated to prepare the solid phase of each group (Table 1).

K_2_HPO_4_ (0.5 M) was added to ultrapure water and stirred with a glass rod until it fully dissolved to form the liquid phase. To cure the mixture, the solid phase powder and the liquid phase were added to a container at a ratio (powder/liquid) of 2.5 g/mL. Figure 1 depicts a schematic representation of the preparation procedure.

In this study, three methods (salting-out, foaming, and polyvinyl alcohol microspheres) were used to make pores in all groups, but only the results of the 10% BG group are discussed.

i. Salting-out: The bone repair material powder, NaCl, and curing solution were uniformly mixed in a certain proportion, and then placed in deionized water for soaking for 48 h after curing, such that NaCl was completely dissolved, and the porous scaffold material was obtained.

ii. Foaming: The bone repair material powder was added with NaHCO_3_, and the curing solution was added with a hydrochloric acid (0.1 mol/L). When the solid phase powder and the liquid phase were mixed the sodium bicarbonate and hydrochloric acid reacted to form a foam. After curing, the porous scaffold material was obtained.

iii. Polyvinyl alcohol microspheres: Soluble crystal particles such as Polyvinyl alcohol are added into the powder, which was placed in water or other solvents to dissolve the crystal particles after the material solidified, such that pores are left, resulting in a large-aperture porous scaffold material.

### 2.2. Characterization of The Materials

#### 2.2.1. Appearance and Surface Morphology Observation

The color, texture, and molding state of the materials were observed under natural light, and photographed and recorded using a digital camera. The materials were completely dried and then made into samples with a diameter of 6 mm and a length of 10 mm. A thin gold layer was coated on the surface and the samples were observed using scanning electron microscopy (SEM) under different magnifications (resolution = 4 nm, 40 A, and the accelerating voltage was 0.3–30 kW) to observe the microscopic morphology of each group of materials. Three samples were selected for observation in each group, and the microstructure and pore distribution and size of the materials were determined.

#### 2.2.2. X-ray Diffractometer Analysis

The structural phases of the samples were analyzed using X-ray diffraction (UIDIA type IV, Boston, MA, USA). Test parameters: Monochromatic Cu-Kα radiation, with a tube voltage of 40 kV, a tube current of 40 mA, a wavelength of 1.5406, ands a scanning speed of 2°/min.

#### 2.2.3. Fourier Transform Infrared Spectroscopy (FTIR) Analysis

The final reaction product after curing of the material was analyzed by FTIR using an IRTracer-100 FTIR instrument (Shimadzu, Kyoto, Japan).

#### 2.2.4. Solidification Time

Referring to the GB/T1346 standard, the composite material samples were mixed and placed in an incubator at 37 °C and 75% humidity. The timer was started when the blending solution was added. The composite material sample was placed in the smart Vicat instrument (LXT-WK300, Shenzhen, China) every minute. When the test needle sank by less than 0.5 mm, the material was regarded as having reached the final setting and the time was recorded.

#### 2.2.5. Injection Performance

For each group of three samples, 4 g of composite solid powder and curing liquid were mixed to form a paste of bone repair material, transferred to a 5 mL syringe, the front air was pushed out, the syringe was installed on a universal material testing machine, and the calibration value X1 of the syringe was recorded. Then, a certain force was exerted on the syringe to squeeze the bone repair material out of the syringe until the force increased to 100 N, at which point the calibration value X2 of the syringe was recorded, and the injection rate of the bone repair material (A) was calculated as: A = (X1 − X2)/X1 × 100%

#### 2.2.6. Anti-Collapse Performance

The material was injected into a petri dish filled with saline, placed in a constant temperature and humidity box (temperature = 37 °C, humidity > 90%), and the morphology of the composite material was observed at 0 and 24 h.

#### 2.2.7. Porosity/Water Absorption

A ceramic density/porosity/water absorption tester (mZ-C150, Shenzhen, China) was used to test the porosity/water absorption of each group according to the American Society for Testing and Materials (ASTM) C20 standard. The percentage of the volume of open pores in a material that can be saturated or filled with water relative to the volume of the material in its natural state was defined as open porosity in the results.

#### 2.2.8. Compressive Strength Test

The compressive strength of the prepared materials was tested using a universal testing machine (Instron, Canton, MA, USA). The samples used for mechanical load testing were prepared with a diameter of 6 mm and a length of 10 mm. Three samples were measured for each group and the values were recorded.

#### 2.2.9. Material Degradation Performance

After weighing the initial material to obtain weight M0, the materials were soaked in Tris-HCL solution for 1, 3, 5, 7, and 14 d, and weighed after drying to obtain weight M1. The percentage of mass loss (w%) was calculated. W% = (M0 − M1)/M0 × 100%.

#### 2.2.10. Microstructure after Degradation

The morphology of the composites was observed under the SEM at the same selected time points as in Section 2.2.1.

### 2.3. Biosafety and Osteogenesis In Vitro Testing

#### 2.3.1. Preparation of the Leachate and Total Elemental Analysis

The cured composites (4 g for each sample) were soaked in 50 mL of alpha-minimal essential medium (α-MEM) medium, placed in a CO_2_ incubator at 37 °C for 24 h, filter sterilized, and adjusted to pH 7.4. Inductively Coupled Plasma-Optical Emission Spectrometry (ICP-OES, ICAP-6300, Thermo Fisher, Scientific, Waltham, MA, USA) was used to determine the concentration of B, Ca, Mg, P, Si, and Sr ions in the leaching solution of the BG composites.

#### 2.3.2. CCK-8 Proliferation Assay

The mouse MC3T3-E1 osteoblast cell line (Shanghai Cell Bank, Chinese Academy of Sciences) was cultured in α-MEM medium containing 10% fetal bovine serum and 1% penicillin-streptomycin. The material leachate was collected and added to the wells of a 96-well plate containing MC3T3-E1 cells. The culture medium was removed on the 1st, 3rd, and 5th day; 100 μL of culture medium and the Cell Counting Kit-8 (CCK8) reagent was added to each well and incubated for 2 h in the dark. The optical density at 450 nm (OD450) was then determined using a microplate reader (ELx800, Biotek, Winooski, VT, USA).

#### 2.3.3. Live/Dead Cell Staining

Calcein Acetoxymethyl ester (AM) (5 μL) and 30 μL of propidium iodide (PI) were added to 10 mL of phosphate buffered saline (PBS) to prepare the staining solution. Then, 200 μL of the staining solution was added to the wells of a 96-well plate in which cells were cultured with the leachate for 1 day, incubated for 30 min in the dark, and then observed using a fluorescence microscope.

#### 2.3.4. Alkaline Phosphatase (ALP) and Alizarin Red S (ARS) Staining

MC3T3-E1 cells were cultured in the leachate liquor with osteogenic medium (OM: α-MEM + 10% FBS + 1% penicillin–streptomycin solution + 1% β-glycerophosphate + 1% ascorbic acid + 0.1% dexamethasone), cultured for 14 and 21 days, and then subjected to ALP and ARS staining. A nitro-blue tetrazolium/5-bromo-4-chloro-3’-indolyphosphate (BCIP/NBT) kit (Beyotime, Beijing, China) was used to perform the ALP staining assay. A 1% solution of ARS was used for staining.

#### 2.3.5. Quantitative Real-Time Reverse Transcription Polymerase Chain Reaction (qRT PCR) Assay

After 7 days of MC3T3-E1 cell culture with the leachate, total RNA was extracted. cDNA synthesis was performed using a reverse transcription kit (Takara, Shiga, Japan). Then, the quantitative real-time PCR (qPCR) step of the qRT-PCR protocol was used to detect *Osx* (encoding Osterix), *Ocn* (encoding Osteocalcin), *Runx2* (encoding RUNX family transcription factor 2), *Opg* (encoding Osteoprotegerin), and *Col1* (encoding Collagen type I) mRNA expression. *Gapdh* (encoding glyceraldehyde-3-phophate-dehydrogenase) was used as an internal reference; the primers used in this study are shown in Table 2.

#### 2.3.6. Statistical Analysis

IBM SPSS 24.0 software (IBM Corp., Armonk, NY, USA) was used to carry out analysis of variance (ANOVA) for statistical analysis. All quantitative data are presented as the mean ± SD. All experiments were repeated three times independently, and *p* < 0.05 was regarded as statistically significant.

## 3. Results

### 3.1. Material Synthesis and Characterization

Different ratios of BG/CSH were added to CP to make composite powders, comprising blending with the solidified liquid, mixing evenly, and filling them into a syringe. Both the Rebone and BG groups were fully extrudable from the syringe. All groups had X2 = 0 and A = 100%, indicating that they were totally injectable. As shown in Figure 2A, the composite material was paste-like. Under natural light, the material was cylindrical after injection, the white surface had a small microporous structure (Figure 2B). SEM observation showed that the surface of 0% BG material was flat, with fine pores scattered on the surface. The voids in the 10% BG material were more numerous than those in the 0% BG group, and the surface of the 20% BG material is not smooth; more pores could be observed even by the naked eye, and the pore size was about 20 μm. The porosity test showed no statistical difference among the groups (*p* > 0.05; Figure 2C).

The characterization of the 10% BG group is shown in Figure 2. X ray diffraction (XRD) showed that the main crystal phases of the modified composite material are hydroxyapatite and calcium sulfate (Figure 2D). The FITR spectrum of the composite material was determined, in which 564 cm^−1^ and 602 cm^−1^ are the P-O stretching vibration absorption peaks of PO_3_^2−^, indicating that the reaction product contains hydroxyapatite; the strong absorption peak at 1033 cm^−1^ is the Si-O asymmetry vibration absorption peak, showing that the reaction product contains silica (the main component of BG); 1094 cm^−1^ is the sulfate stretching vibration absorption peak, and 3434 cm^−1^ is the H-O vibration absorption peak, which means that the reaction product contains calcium sulfate dihydrate.

The phase results of FITR and XRD (Figure 2E) crystallography revealed that the BG + CSH bone repair material underwent a multi-curing reaction, and the final reactants were hydroxyapatite, bioactive glass, and calcium sulfate dihydrate. The reaction equation was as follows
CaSO_4_•1/2H_2_O + 3/2H_2_O → CaSO_4_•2H_2_O
2Ca_4_(PO_4_)_2_O + 2CaHPO_4_•2H_2_O→ Ca10(PO_4_)_6_(OH)_2_ + 4H_2_O

The solidification time of different ratios BG/CSH groups was about 12 min (Figure 2F), and there was no significant difference among the groups (*p* > 0.05).

### 3.2. Anti-Collapsibility, Porosity, and Degradation

The morphology of the composite material injected into physiological saline is shown in Figure 3A. After 24 h, the composite material retained its original shape without collapse, and the soaking solution was clear without turbidity.

To test the porosity of the composite material, we used an MZ-C150 ceramic density/porosity/water absorption tester together with the ASTM C20 standard. The total porosity of the 10% BG group of the material prepared by the foaming method reached 56%, and the total porosity of the 20% BG group reached 55% (Figure 3B). 

The mechanical strength was 13.4 ± 3.1 MPa for 0% BG + 50% CSH, 15 ± 1.6 MPa for 10% BG + 40% CSH, and 18.8 ± 1.5 MPa MPa for 20% BG + 30% CSH; in terms of mechanical properties, the strength of Rebone is around 30 Mpa. With the addition of calcium sulfate, the strength of the material decreased significantly. As the content of BG in the composite increased, the compressive strength increased, and the compressive strength of the 20% BG group reached more than 15 Mpa, (Figure 3C,D). Figure 3E shows that the surface pore size increased to approximately 50 μm.

In Tris-HCl solution, the material degraded slowly. The degradation curves of Rebone and the 0% BG groups were the same, with a degradation rate of 5% after 28 days. The degradation rate of the BG groups in Tris-HCl increased significantly after 14 days, and the degradation rate was greater than 20% at 28 days, in which the degradation rate of the 20% BG group was close to 25% (Figure 3F). The SEM results showed that the surface pores of the material gradually enlarged after 3 days of degradation. At 14 days, the surface pores of the BG composite reached 50–200 μm in diameter (Figure 3G). In addition, the open porosity of the material increased by approximately 5–10% after 14 days.

### 3.3. Material Biosafety Performance Testing

By detecting the ionic composition and concentration of the leachate, we determined the contents of B, Ca, Mg, Si, P, and Sr (Table 3). In the 10% BG and 20% BG groups, the levels of the ions increased significantly compared with those in the 0% group. Among them, the Si content increased with increasing BG concentration (Table 3).

The live/dead cell assay showed that there were more red-stained (dead) cells in the 30% BG group. Except in the 30% BG group, the cells were mainly green-stained live cells, which were evenly and clearly distributed, and there were relatively few red-stained cells (Figure 4A). CCK-8 detection showed that with the increase in culture time, the OD value of each group of materials gradually increased, i.e., the cells could proliferate well (Figure 4B).

### 3.4. Osteogenic Effect Detection

After culturing MC3T3-E1 cells with the leachates of the different groups for 14 days and 21 days, ALP and ARS staining showed that all materials had obvious positive staining. The ALP and ARS staining deepened as the content of BG increased. The 20% BG group had the deepest staining, and the quantitative ALP and ARS results showed that 20% BG had the most obvious effect on mineralization in vitro (Figure 4C,D).

### 3.5. Detection of Osteogenic Gene Expression In Vitro

The mRNA expression levels of *Osx*, *Ocn*, *Runx2*, *Opg*, and *Col1* were detected using qRT-PCR. Among the groups, the 20% of BG group had the highest expression of *Osx*, *Ocn*, *Runx2*, and *Col1* mRNA (*p* < 0.05). However, there was no significant difference for *Opg* (Figure 4E).

## 4. Discussion

In the treatment of maxillofacial bone defects, especially alveolar crest defects in dentistry, bone repair materials consistent with that of the bone defects would achieve effective local space maintenance and subsequent bone repair effect. In clinical practice, injectable self-curing bone repair material can mold to the required shape and form to repair irregular bone defects [11]. At present, there is no ideal injectable self-curing bone repair material. Calcium phosphate or calcium sulfate injectable bone repair materials have limitations, e.g., calcium phosphate is difficult to degrade and is barely substituted by new bone tissue, while calcium sulfate degrades too fast and does not match the growth rate of new bone tissue. Therefore, modification of injectable bone repair materials is necessary to meet the requirements of bone defect repair [12]. In addition, the porosity of injectable self-curing bone repair materials is about 30% and their pore size is less than 10 microns. This makes it difficult for tissue fluid and cells to penetrate the material [11]. Previous research indicated that cells can grow into the material when the pore size is 100 μm or larger, which also benefits the vascularization of the materials and ensures the supply of nutrients to the internal tissues [13].

Therefore, an ideal self-curing bone repair material should not only have self-curing properties, but also have a good cell interface and a three-dimensional porous structure. The following features should be present [14]:(1)Good biocompatibility to avoid immune reactions in the implanted area;(2)Good biodegradability to promote bone remodeling;(3)Porosity for cell adhesion and nutrient transport;(4)Moderate mechanical strength to support defect remodeling;(5)Bone conductivity and osteoinduction.

In this study, the composite of calcium phosphate and calcium sulfate was mixed with a certain proportion of BG. Three-phase injectable materials with different BG proportions were designed and synthesized, consisting of calcium phosphate/calcium sulfate/BG. XRD and FTIR verified the composition and crystalline structure of the composites.

It is believed that a curing time of 15–20 min is suitable for bone repair [15]. The materials in this study had an average solidification time of about 12 min, making them suitable. In addition, after BG and CSH were added to CP, the compressive strength decreased significantly to 15–20 Mpa, which is lower than that of cortical bone (170–193 Mpa), but a little higher than that of trabecular bone (2–12 Mpa) [16], which can meet the clinical requirements for non-load bearing areas, such as maxillofacial bone defects. Meanwhile, it is desirable that injectable bone repair materials should remain stable for 24 h for space maintenance [17,18], which is important to avoid washing away by tissue fluid. In this study, the surface of the composite materials had a uniform shape after injection and remained in their original state without collapsing after soaking in physiological saline for 24 h, with a clear leaching solution without turbidity, which indicated that the composite materials had good anti-collapsibility characteristics.

Previous studies found that the optimal porosity for bone repair materials should be greater than 50%, and the pore size should be between 100 and 500 μm [13]. Therefore, we compared three methods of pore formation and found that the foaming method was suitable for these self-curing bone repair materials. The pore diameter resulting from the foaming method was about 50 μm. After 14 days of degradation, the pore diameter reached 50~200 μm. SEM analysis showed that the pore diameter gradually increased with time. Yangzi et al. found that the phosphate solubility of a phosphate/sulfate composite was the lowest in liquid at pH 8, and the degradation was mainly that of calcium sulfate. When calcium sulfate was completely degraded, bone repair materials would form a porous structure containing only calcium phosphate groups. Such a porous structure provides spaces for blood vessel formation and bone cell migration [19]. Meanwhile, the communication structure between pores can promote the migration of bone cells and the exchange of nutrients [20]. At the same time, the phagocytosis of cells in vivo will accelerate the degradation of the composite materials, and the released calcium ions and phosphate ions, which can activate platelets to release bone morphogenetic proteins and platelet-derived growth factors that stimulate the proliferation and osteogenic differentiation of mesenchymal stem cells, such that the composite materials are gradually replaced by bone tissue [21].

The porosity among the BG groups was not significantly different, but was higher than that in the control group, which indicated that the different proportions of BG had no effect on viscosity, whereas the modification endowed higher porosity to the composite materials.

In this study, the BG/CSH-modified bone repair material was degraded under the action of Tris-HCl and the degradation rate reached about 20% over 14 to 28 days. For bone repair materials, a suitable degradation rate should form macropores of about 100 μm in 2–4 weeks. It is considered that a better osteogenic effect can be achieved when the degradation rate matches the rate of new bone formation [22].

Combined with the results of material shaping and compressive strength, the degradation results showed that the composite material had good shaping ability and strength during injection molding. When immersed in body fluid, the pores of the material gradually expand and communicate with each other, which provides a suitable scaffold structure and cell growth conditions for bone repair similar to other partially degraded materials [23].

The results of the present study showed that the levels of Si, Ca, Mg, and B ions in the BG group were significantly higher than those in the control group as determined by analysis of the components of the leaching solution. Si ions released during the degradation process are beneficial to angiogenesis and osteogenesis [24]. A gel layer rich in SiO_2_ can form because Ca^2+^ and PO_4_^3−^ are enriched by BG. Moreover, another layer containing carbonated hydroxyapatite (HA) bonds with human bones and mineralizes to form new bones [25]. Studies have shown that an appropriate amount of Si ions can markedly inhibit osteoclasts and activate osteoblasts in vitro, via a mechanism in which Si inhibits the activated inflammatory mitogen-activated protein kinase (MAPK) and nuclear factor kappa B (NF-κB) signaling pathway, which promotes the caspase-dependent apoptosis of macrophages [26]. The results of cell proliferation tests and biological safety tests (CCK-8 and Calcein-AM/PI staining) showed that cell proliferation was maintained when the composition comprised 10% BG and 20% BG. Therefore, the composite material had no effect on cell proliferation (Figure 4A). Thus, the material has high biocompatibility and low toxicity, allowing further study of the osteogenesis effect of the composite materials in vitro [27].

The increased ALP activity and ARS staining indicated significant active osteoblasts performance in the 20% BG group, suggesting that the 20% BG group had higher osteogenic activity. Even the 10% BG group showed an increasing trend, but without statistical significance.

The results of qRT-PCR showed that the expression of *Osx*, *Ocn*, *Runx2*, *Opg*, and *Col1* mRNA in osteoblasts was significantly increased under the influence of the composite materials, and the expression levels of *Osx*, *Ocn*, *Runx2*, and *Col1* in the 20% BG group were higher than those in the other groups. Therefore, the CP/CSH/BG (especially 20% BG) could promote the osteogenic activity of MC3T3-E1 cells. Their role in promoting an osteoblast phenotype indicates that the composites have the potential to promote bone formation in vivo.

According to the results of CP/CSH/BG on osteogenesis in vitro, the biological effects of the composite materials verified that the addition of BG and CSH could enhance osteoblast-mediated osteogenesis.

## 5. Conclusions

In summary, based on the limited results of this study, we designed a bone repair material that can achieve self-curing, intraoperative shaping, and postoperative space maintenance.

By adjusting the proportion of BG, CSH, and CP in the composite material, the degradation rate and biocompatibility of the material can be modulated. The material continuously releases Si, Ca, and P into plasma, and the sediment can promote osteogenesis of osteoblasts in vitro. This study provides strategies to improve injectable bone repair materials and introduces a new injectable self-curing material modified by BG/CSH, which might have application potential in bone regeneration.

## Figures and Tables

**Figure 1 materials-15-07898-f001:**
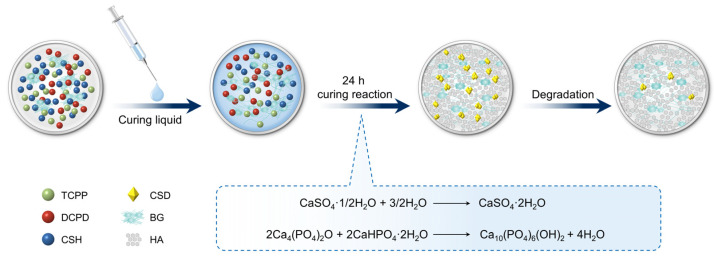
Schematic diagram of the preparation of the self-curing composite material. TTCP, tetracalcium phosphate; DCPD, dicalcium phosphate; BG, bioactive glass; CSH, calcium sulfate hemihydrate; CSD, calcium sulfate dihydrate, HA, hydroxyapatite.

**Figure 2 materials-15-07898-f002:**
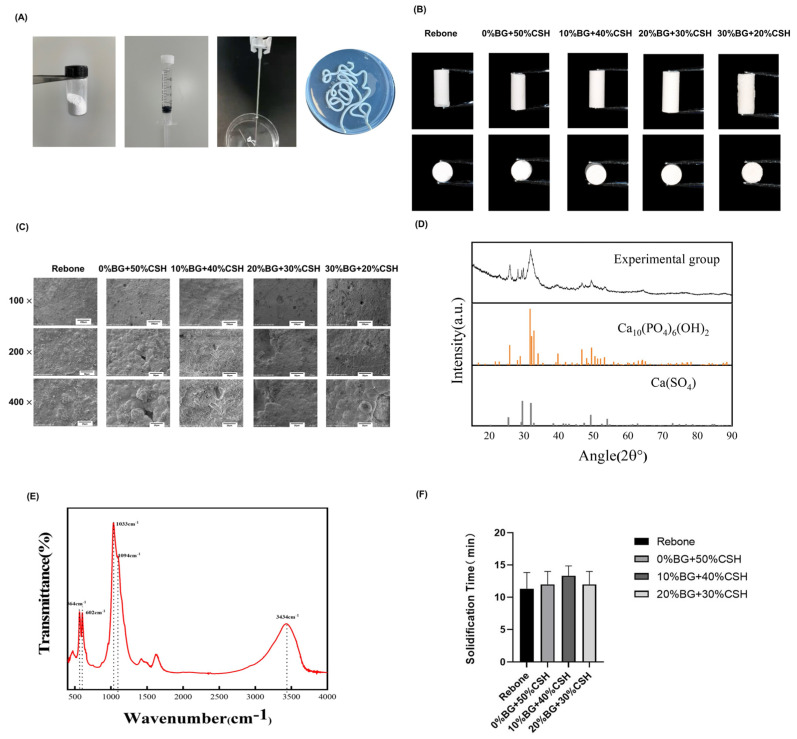
Material synthesis and characterization of the composite materials (**A**) The solid powder and the curing liquid were added into a container at 2.5 g/mL, blended manually for 1 min, and introduced into a ring handle syringe for later use; (**B**) the composite material after curing; (**C**) image of the surface of the initial self-cured composite under the electron microscope; (**D**) X−ray diffraction (XRD) pattern of the composite; (**E**) Fourier transform infrared spectroscopy (FITR) pattern of the composite; (**F**) solidification time analysis.

**Figure 3 materials-15-07898-f003:**
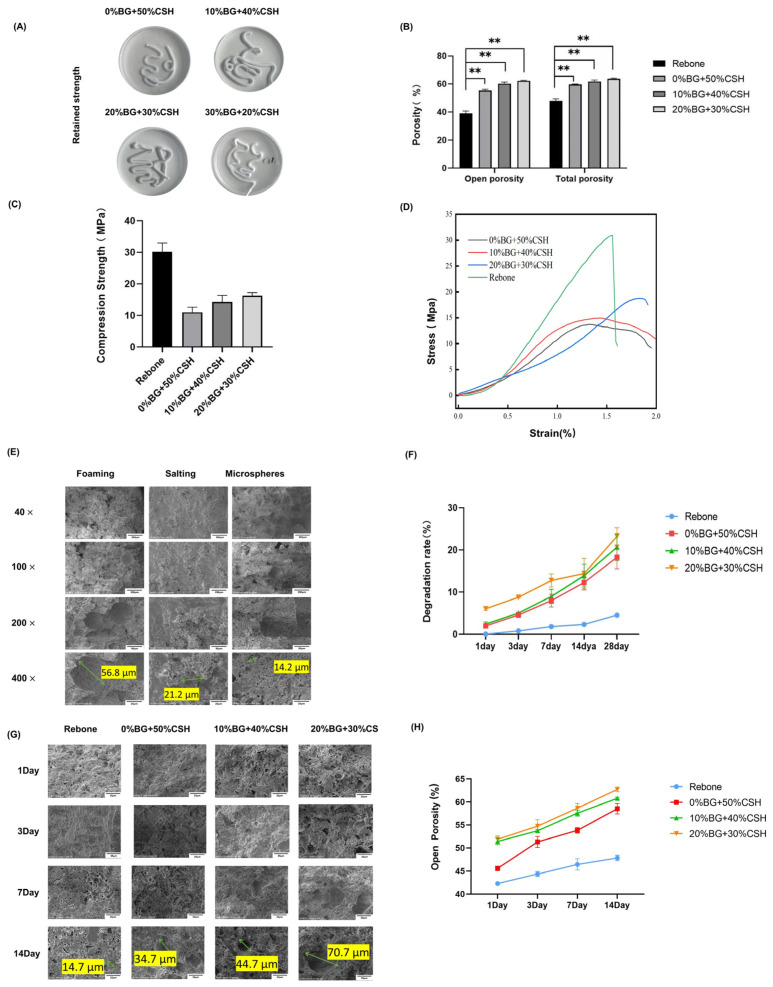
The anti-collapsibility, porosity, and degradation properties of the composite materials. (**A**) Anti-collapse properties of the self-cured material; (**B**) composite pore size statistics(** means statistically significant, *p* < 0.01.); (**C**) comparison of the compressive strength of the composite materials; (**D**) stress–strain curves; (**E**) scanning electron microscopy (SEM) images of the composite materials after three pore-making methods; (**F**) graph showing the Tris-HCl dissolution of the composite materials; (**G**) SEM image of the composite materials after degradation (1000× magnification); (**H**) open porosity for 1–14 days after degradation.

**Figure 4 materials-15-07898-f004:**
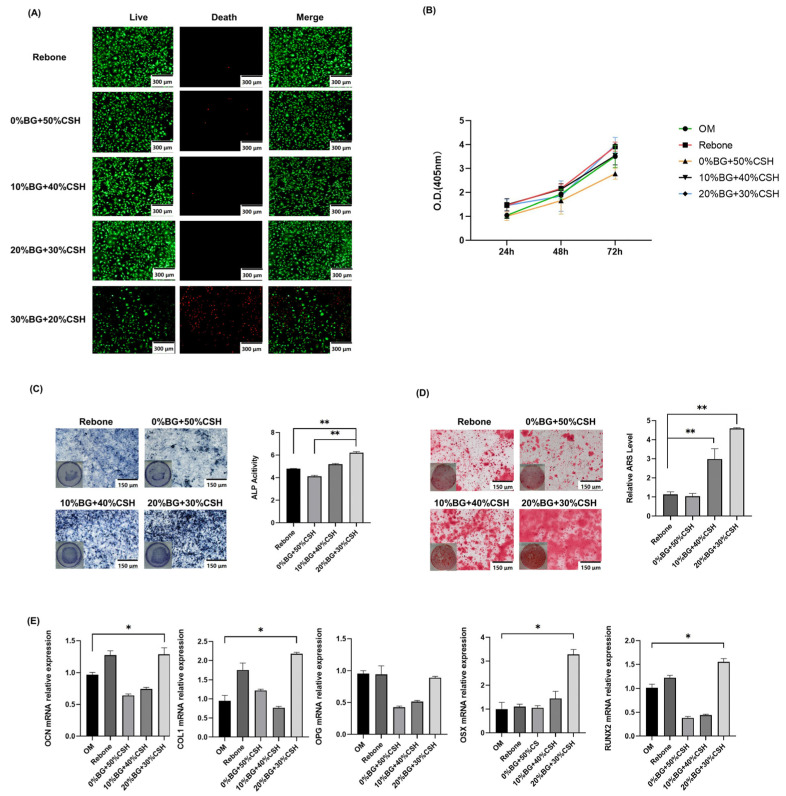
Biosafety test and in vitro osteogenesis tests of the composite materials. (**A**) Experimental results of Calcein-acetoxymethyl (AM)/propidium iodide (PI) staining (live/dead staining); (**B**) Cell Counting Kit-8 (CCK8) detection of cell proliferation; (**C**,**D**) qualitative and quantitative analysis of alkaline phosphatase (ALP)/Alizarin red S (ARS) staining; (**E**) results of qRT-PCR (OM, osteogenic medium, * means *p* < 0.05, ** means *p* < 0.01).

**Table 1 materials-15-07898-t001:** Material components and groups.

	Group	Component
1	Control	Rebone (Commercial CP)
2	0% BG	0% BG + 50% CSH + 50% CP
3	10% BG	10% BG + 40% CSH + 50% CP
4	20% BG	20% BG + 30%CSH + 50%CP
5	30% BG	30% BG + 20% CSH + 50% CP

CP, calcium phosphate; BG, bioactive glass; CSH, calcium sulfate hemihydrate.

**Table 2 materials-15-07898-t002:** PCR Primers.

Gene	Forward	Reverse	Species
*Gapdh*	CGACAGTCAGCCGCATCTT	CCAATACGACCAAATCCGTTG	Mouse
*Runx2*	GATGACACTGCCACCTCTGAC	GGGATGAAATGCTTGGGAAC	Mouse
*Col1*	GATGGAGAACTGTCATGGGAAC	GGAGGATCATAGTGAGAGCGG	Mouse
*Ocn*	AGCAAAGGTGCAGCCTTTGT	GCGCCTGGGTCTCTTCACT	Mouse
*Osx*	CCTCCTCAGCTCACCTTCTC	GTTGGGAGCCCAAATAGAAA	Mouse
*Opg*	ACCCAGAAACTGGTCATCAGC	CTGCAATACACACACTCATCACT	Mouse

**Table 3 materials-15-07898-t003:** Active ion concentration of leaching liquors (mg/L).

	B	Ca	Mg	P	Si	Sr
Rebone	0.57	39	7.05	42	8.07	0.04
0% BG	0.31	52	15.15	28	0.59	0.02
10% BG	4.04	557	102.6	2.21	18.73	0.73
20% BG	3.54	368	64.74	1.02	47.61	0.4

## Data Availability

Not applicable.

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
