# Peer review of "Structure and Properties of Bioactive Glass-Modified Calcium Phosphate/Calcium Sulfate Biphasic Porous Self-Curing Bone Repair Materials and Preliminary Research on Their Osteogenic Effect"

_materials, 2022, doi:10.3390/ma15227898_

Round 1

Reviewer 1 Report

The present work is aimed at the study of composite materials for the repair of bone defects. Calcium phosphates, calcium sulphate and bioactive glass were mixed to obtain an injectable material.

Introduction

- Please explain more about the novelty of the proposed work, since the combination of calcium phosphate and sulphate based cements with bioactive glasses is already proposed in the literature.

Materials and Methods

- What is the liquid to powder ratio (g/mL) used for preparation and cements? In the text, only the mixing rate is provided ('The solid phase powder and the liquid phase have been added to a container at the rate of 2.5 g/ml' - Line 100)

-Three different methods (salting, foaming and polyvinyl alcohol microspheres) were used to induce a porosity in the sample with 10% BG:

Why have not all samples been tested?

Was it checked that NaCl (method i), NaHCO3 and HCl (method ii) or polyvinyl alcohol microspheres (method iii) did not remain inside the material?

How was the porosity of the other samples (other than 10% BG) obtained?

Results

- In general, the 30% BG sample was presented together with the other compositions (group n.5 Table 1), but was not fully characterized as the others. In fact, many results are missing: porosity, mechanical force in compression, the rate of degradation in Tris-HCl, post-degradation test porosity, active  ion concentrations of leaching liquors (mg/L), CCK-8 detection and osteogenic effect, detection of osteogenic gene expression in vitro.

Is this due to the results obtained in the biosafety test and in vitro osteogenesis tests?

-LINE 293. ‘Figure 3D shows that the surface porosity of the material was greatly improved and the surface pore size increased to 50μm’. The phrase is not clear: please explain 'greatly improved' by what.

- In Figure 3D and 3F the scale bar is not clearly visible.

- Figure 3D: please explain the difference between ‘foaming L’ and ‘foaming H’

-LINE 376-378: please provide appropriate reference for the proposed compression force ranges.

- Compressive force: it is desirable to insert the stress-strain curves of the samples obtained, which give an indication of the toughness of the material.

Author Response

Point 1: Please explain more about the novelty of the proposed work, since the combination of calcium phosphate and sulphate based cements with bioactive glasses is already proposed in the literature

Response 1Thanks for recommends. The use of a composite graft containing CS and CP has been described in several reports and studies. However, the current materials are lack of proper porosity for cell migration and new tissue formation. The introduction has been rewritten for more clearly statement about it.

Point 2: What is the liquid to powder ratio (g/mL) used for preparation and cements? In the text, only the mixing rate is provided ('The solid phase powder and the liquid phase have been added to a container at the rate of 2.5 g/ml' - Line 100)

Response 2: We apologize for the mistake in using of “rate”, correct word is “ratio”. The word has been revised in Line 75 Page 2

Point 3.1Three different methods (salting, foaming and polyvinyl alcohol microspheres) were used to induce a porosity in the sample with 10% BG: Why have not all samples been tested? How was the porosity of the other samples (other than 10% BG) obtained?

Response 3.1: All groups were treated by poring and the results were consistent between each group, thus only the 10%BG group was included in the article because of limit of article length. The results of other groups were submitted in this respose as supplementary data.(Supplementary Figure 1 ABC). The statement of the restricted data was added in the article in line 81 

Point 3.2Was it checked that NaCl (method i), NaHCO3 and HCl (method ii) or polyvinyl alcohol microspheres (method iii) did not remain inside the material?

Response 3.2: The PH was checked to make sure there was not residual HCl. The PH of the sample leach solution (see Supplement for PH results following). The results show that the ph of samples in all groups were slightly alkaline. NaHCO3 is used little more than HCl to ensure that HCl could reacted completely. The trace residual NaHCO3 can be buffered easily by the buffer system of body fluid. We did not check residual NaCl and Polyvinyl alcohol microspheres(PVA), because that NaCl  is a common ingredient of body fluid which can be buffered easily in vivo. And PVA is a biocompatible semi-crystalline synthetic polymer that has been used in the field of biotechnology. PVA is biodegradable and non-toxic(ung Giu Jin. Production and Application of Biomaterials Based on Polyvinyl alcohol (PVA) as Wound Dressing.J. Chem Asian . 2022 Sep 6;e202200595. doi: 10.1002/asia.202200595.)

Point 4In general, the 30% BG sample was presented together with the other compositions (group n.5 Table 1), but was not fully characterized as the others. In fact, many results are missing: porosity, mechanical force in compression, the rate of degradation in Tris-HCl, post-degradation test porosity, active  ion concentrations of leaching liquors (mg/L), CCK-8 detection and osteogenic effect, detection of osteogenic gene expression in vitro.

Is this due to the results obtained in the biosafety test and in vitro osteogenesis tests?

Response 4Sorry for unsatisfied results presentation. The 30% BG samples presented cytotoxic characters in the biosafety test. Thus the 30% BG samples were abandoned in the following other tests. It has been revised in Line 276-277

Point 5-LINE 293. ‘Figure 3D shows that the surface porosity of the material was greatly improved and the surface pore size increased to 50μm’. The phrase is not clear: please explain 'greatly improved' by what.

Response 5I'm sorry for my misnomer of “greatly improved”, and thanks for your correction, the sentence has been revised as “Figure 3E shows that the surface porosity of the material was about 50μm. in Line 247

Point 6Figure 3D: please explain the difference between ‘foaming L’ and ‘foaming H’

Response 6L and H indicate  different  hydrochloric acid concentration as 0.1mol/L(Low) and 0.2mol/L (High) seperately for foaming method. Considering about the result of 0.1mol/L(Low) shows demanded porosity and the high concentration of 0.2mol/L hydrochloric acid is not necessary for the research aim, so the results of the 0.2mol/L (H) have been exampted from the article. Sorry for unclear describing here. This was revised in materials and methods in Line 87. And the result of Foaming(H) was deleted in Figure 3 E.

Point 7-LINE 376-378: please provide appropriate reference for the proposed compression force ranges.

Response 7The cancellous bone have a compressive strength about  5-10Mpa and  the cortical bone was about 162.2MPa

References :Ricci, J.L, Weiner, M.J. et al. Calcium Sulphate: Bioceramics and their Clinical Application. 2008,Pages 310

Point 8- Compressive force: it is desirable to insert the stress-strain curves of the samples obtained, which give an indication of the toughness of the material.

Response 8Thank you for your remind and indication, the stress-strain curves of each group materials were added in Figure 3-D.

Reviewer 2 Report

Page 2 lines 49-50: Although the authors state that "artificial bone repair materials is unlimited" this statement is only true for autologous bone. However, this availability is common for allografts and xenografts (both come from banks of human or animal material). This sentence should be rewritten.

Page 2 lines 53-54, page 12 lines 400-402, page 13 line 420-422: This statement is false. To produce bone, an osteoid matrix is initially required, which is subsequently mineralized with ions from systemic, not local, metabolism. It should be corrected. The authors should remember that mineralization depends on the generation of the osteoid matrix secreted by the osteoblasts, and that the accumulation of ions alone does not generate bone, only a precipitate.

Page 3, lines 95-96: The authors used CP as a base along with an increasing concentration of BG (0%-30%) and a decreasing concentration of CSH (50%-20%). In the text, however, BG appears from 10% to 30%. This should be corrected.

Page 3: line 102 and subsequent: The authors employ 5 groups of compounds and apparently 3 methods to create porosity (salt-out, foaming and polyvinyl microspheres). In total 8 possibilities and 15 groups. If this is correct, it should be clearly stated in material and methods.

Page 12, line 398 and page 13 line 415 and 420-422: It is generally accepted that the main difference between synthetic and natural materials is the interconnection of the pores (in addition to their size). Natural materials are interconnected while synthetics are rarely connected. Can the authors confirm that in their study this pore connection is real? If so, how did they verify this connection?

Author Response

Point 1: Page 2 lines 49-50: Although the authors state that "artificial bone repair materials is unlimited" this statement is only true for autologous bone. However, this availability is common for allografts and xenografts (both come from banks of human or animal material). This sentence should be rewritten.

Response 1We apologize for the improper statement about the property of artifacial bone repair materials and thanks for instruction about it. The sentence has already been rewritten as  “artificial bone repair materials can be manufactured as demanded structure and volume” .

Point 2: Page 2 lines 53-54, page 12 lines 400-402, page 13 line 420-422: This statement is false. To produce bone, an osteoid matrix is initially required, which is subsequently mineralized with ions from systemic, not local, metabolism. It should be corrected. The authors should remember that mineralization depends on the generation of the osteoid matrix secreted by the osteoblasts, and that the accumulation of ions alone does not generate bone, only a precipitate.

Response 2: We apologize for the mistakes here, and thanks for your instruction. All of the incorrect statements has been revised in Line 345-350 and Line 366-369 ,  related reference is uploaded.

Point 3Page 3, lines 95-96: The authors used CP as a base along with an increasing concentration of BG (0%-30%) and a decreasing concentration of CSH (50%-20%). In the text, however, BG appears from 10% to 30%. This should be corrected.

Response 3Sorry for the error here. In the experimental group, BG should be 0%-30%, CSH is 50%-20%, all these errors have been corrected in Line 71

Point 4Page 3: line 102 and subsequent: The authors employ 5 groups of compounds and apparently 3 methods to create porosity (salt-out, foaming and polyvinyl microspheres). In total 8 possibilities and 15 groups. If this is correct, it should be clearly stated in material and methods.

Response 4Yes, in our research, there are 3 pore-forming methods, 5 groups, so it should be 15 groups of graphics. And the 30% BG samples presented cytotoxic characters in the biosafety test. So the 30% BG samples were abandoned in the following tests. The other samples were tested, because that the results were basically consistent in each group. Only the 10%BG group was included in the article, and the results of other groups were submitted as following supplementary data in this response (Supplementary Figure1 ABC). ). The statement of the restricted data was added in the article in p.2.l.82 

Point 5Page 12, line 398 and page 13 line 415 and 420-422: It is generally accepted that the main difference between synthetic and natural materials is the interconnection of the pores (in addition to their size). Natural materials are interconnected while synthetics are rarely connected. Can the authors confirm that in their study this pore connection is real? If so, how did they verify this connection?

Response 5Thanks for instructions. For sure the porosity especialy interconnected pores play important roles for bone repair materials. There are many methods to test this character. In this study, we tested the open porosity of the material which is recommended by ASTM C20 standard(Standard Test Methods for Apparent Porosity, Water Absorption, Apparent Specific Gravity, and Bulk Density of Burned Refractory Brick and Shapes by Boiling Water). It is issued by ASTM Committee in 2015. In the  method, the water soaking ability of material is tested to represent the open porosity. The revisement was added in the materials and methods in page 7 line 255. the result of it was added in Fig.3H

Reviewer 3 Report

The present manuscript entitled "Structure and properties of bioative glass-modified calcium phosphate/calcium sulfate biphasic porous self-curing bone repair materials and preliminary research on their osteogenic effect" characterizes a mixture of different mineral bone substitute materials in vitro. 

In principle, such work is to be welcomed, since there have not yet been any really resounding successes in this field. 

At the same time, the authors' results are quite promising. 

Nevertheless, the manuscript has weaknesses in large parts that make publication possible only after intensive corrections. 

Relevant weaknesses are listed below: 

Introduction

The introduction shows inaccuracies in the formulations and the current state of knowledge. It also shows a significant deficiency in the literature citations used. Almost 90% of the cited literature are older than 7 years. Here an actualization must absolutely take place. A complete revision also of the contents is necessary. Especially the differentiation between different areas of application should be addressed. Use in the oral space requires different properties than use on large tubular bones.

p.2.l.40: one serious problem are "donor side morbities" like pain, infection etc. p

p.2.l.44: Why is the use of allogenic bone inconvenient?

p.2,l47: what means stable position? With a paste it is also difficult!

p.2l.47: Especially DBM shows no antigenicity (see Söhling et al.). 

Materials and Methods

p.3.l112: why did you use only 10% BG for poring?

p.3.l117: Did you measure the PH after preparation? Was it neutral? Were equivalent amounts of NHCO3 and HCl used?

p.3.l121: What about toxicity in this procedure?

Figure 1: I don't understand the figure under the formula!

p.4l.177: What kind of scaffold/structure/shape was used for mechanical load testing?

Figure 2: Labeling is not consistent (font)

Discussion

Current literature!!!!!!!!!!

p11l.343: Specify what kind of bone repair material you test. for which purpose? Mainly for maxilla facial surgery, dentistry!

p.12l381: Revise with current literature.

p12.l407: What about interconnectivity of the pores? Did you measure it? Please specify! Please 

p.13l415 Where does the information come from that interconnectivity increases due to degradation?

Author Response

Point 1: The introduction shows inaccuracies in the formulations and the current state of knowledge. It also shows a significant deficiency in the literature citations used. Almost 90% of the cited literature are older than 7 years. Here an actualization must absolutely take place. A complete revision also of the contents is necessary. Especially the differentiation between different areas of application should be addressed. Use in the oral space requires different properties than use on large tubular bones. 

Response 1Thanks for pointing out the mistakes. I have consulted the literature in recent years, and rewrote the introduction and inserted new literatures.

Point 2: p.2.l.40: one serious problem are "donor side morbities" like pain, infection etc.

p.2.l.44: Why is the use of allogenic bone inconvenient?

p.2,l47: what means stable position? With a paste it is also difficult!

Response 2:

p.2.l.40: one serious problem are "donor side morbities" like pain, infection etc.

Thanks for instruction, the sentence has been revised in p.1.l.41

p.2.l.44: Why is the use of allogenic bone inconvenient?

 Sorry for unclear statement. The inconvenient of allogenic bone means that the shaping of it as demanded is time-consumpted and technically sensitive. The sentence has been revised under instruction in p.2. l.42-44.

p.2,l47: what means stable position? With a paste it is also difficult!

Sorry for unclear statement and mistakes. The sentences have been revised under this instrction in p.2 p.45-48.

p.2l.47: Especially DBM shows no antigenicity (see Söhling et al.)

The article of Söhling et al(Materials 2020, 13, 3120) has been studied. Thanks for instruction and sorry for improper statement. This error statement has been erased.

Point 3p.3.l112: why did you use only 10% BG for poring?

Response 3All groups were treated by poring and the results were consistent between each group, thus only the 10%BG group was included in the article because of limit of article length. The results of other groups were submitted in this respose as supplementary data.(Supplementary Figure 1 ABC). The statement of the restricted data was added in the article in p.2.l.82  

Point 4p.3.l117: Did you measure the PH after preparation? Was it neutral? Were equivalent amounts of NHCO3 and HCl used?

Response 4 Yes we did. The PH of the sample leach solution (see Supplement for PH results) and the overall PH results is following in Supplementary  Figure 2 in this response. The results show that the ph of samples in all groups were slightly alkaline. NaHCO3 is used little more than HCl to ensure that HCl could reacted completely. The trace residual NaHCO3 can be buffered easily by the buffer system of body fluid.

Point 5: p.3.l121: What about toxicity in this procedure?

Response 5Thanks for reminding about the biocompatibility of Polyvinyl alcohol microspheres(PVA). It is a biocompatible semi-crystalline synthetic polymer that has been used in the field of biotechnology. PVA is biodegradable and non-toxic, the reference is presented below.

References: Sung Giu Jin. Production and Application of Biomaterials Based on Polyvinyl alcohol (PVA) as Wound Dressing.J. Chem Asian . 2022 Sep 6;e202200595. doi: 10.1002/asia.202200595.

Point 6Figure 1: I don't understand the figure under the formula!

Response 6Sorry for bad presentation of schematic diagram. We revised Figure1 and change the symbols into more readable ones.

Point 7p.4l.177: What kind of scaffold/structure/shape was used for mechanical load testing?

Response 7Sorry for the unclear statement. The samples used for mechanical load testing were prepared as 6mm in diameter and  10mm in length, the information has been added in the materials and methods. see p.4.l.144

Point 8Figure 2: Labeling is not consistent (font)

Response 8Sorry for inconsistent labeling. The labeling in figure 2 has been revised. 

Point9p11l.343: Specify what kind of bone repair material you test. for which purpose? Mainly for maxilla facial surgery, dentistry!

Response 9Sorry for unclear statement and thanks for instructions. The sentence has been rewrited in p.10 l.296

 Point 10p.12l381: Revise with current literature.

Response 10Thanks for instructions about the literature citations, more recent literatures have been cited here. 

Point 11&12p12.l407: What about interconnectivity of the pores? Did you measure it? Please specify! p.13l415 Where does the information come from that interconnectivity increases due to degradation?

 Response 11&12Sorry for unclear statement. The open porosity was test to represent the internal connectivity and the results was added in Figure 3B.H according to the ASTM C20, the reference has been added in materials and mechods. See p.6. line.239

Round 2

Reviewer 2 Report

The authors have improved their study and responded to the suggestions made. I consider the study suitable for publication.

Author Response

point:The authors have improved their study and responded to the suggestions made. I consider the study suitable for publication.

response:Thank you for your advice!Hve a nice day!

Reviewer 3 Report

The new version is a substantial improvement. Some corrections still need to be made before publication: 

The manuscript needs to be revised grammatically and textually. Especially the new passages have serious flaws, e.g.: 

p.1 l.38 "Among them, autogenous bone has the best therapeutic effect but complex in manipulation for bone..."

p.1 l.42 "Allogeneic and xenogeneic materials, has a natural micropore structure 42 that is conducive to osteogenesis..."

p.2 l.57 "...used to synthesize injectable 57 composite bone substitute materials. And different proportions..."

etc.

_______________________________________________________________

Please provide a picture of a test body. (p.4 l.14 )

___________________________________________________________________

Figure 3H => What is the unit of porosity?`Define the term "open porosity"!

Author Response

Point 1

 p.1 l.38 "Among them, autogenous bone has the best therapeutic effect but complex in manipulation for bone..."

 p.1 l.42 "Allogeneic and xenogeneic materials, has a natural micropore structure 42 that is conducive to osteogenesis..."

 p.2 l.57 "...used to synthesize injectable 57 composite bone substitute materials. And different proportions..."  etc.   

Response 1

  1. 1 l.21 “CP/CSH/BG was almost as strong as trabecular bone…”

   The sentence has been revised under instruction in p.1 l.22

  1. p.1 l.38 "Among them, autogenous bone has the best therapeutic effect but complex in manipulation for bone..."

   The sentence has been revised under instruction in p.1 l.39

  1. p.1 l.42 "Allogeneic and xenogeneic materials, has a natural micropore structure 42 that is conducive to osteogenesis..."

   The sentence has been revised under instruction in p.1 l.43

  1. p.2 l.57 "...used to synthesize injectable composite bone substitute materials. And different proportions..."

   The sentence has been revised under instruction p.2 l.57-60

  1. We checked the grammar and spelling errors and corrected them at p.11.363 p.12 l. 385 390--392.

Point 2: Please provide a picture of a test body. (p.4 l.14 ) 

Response 2: The samples were prepared as 6mm in diameter and 10mm in length(Supplementary Figure3 A.B)

An ceramic density/porosity/water absorption tester (mZ-C150, Shenzhen, China) was used to test the porosity/water absorption of each group according to the American Society for Testing and Materials (ASTM) C20 standard.( Supplementary Figure3 C)

Point 3Figure 3H => What is the unit of porosity?`Define the term "open porosity"!

Response 3: Sorry for missing mark in 3H, the unit of porosity here is percentage(%), the unit of porosity was added in Figure 3H

Open porosity refers to the percentage of the volume of open pores in a material that can be saturated or filled with water relative to the volume of the material in its natural state. The definition was added in Materials and Methods in p.4 l.143
